# Directions of Changes in the Content of Selected Macro- and Micronutrients of Kale, Rutabaga, Green and Purple Cauliflower Due to Hydrothermal Treatment

Joanna Kapusta-Duch [1],*, Adam Florkiewicz [2], Teresa Leszczyńska [1] and Barbara Borczak [1]

[1] Department of Human Nutrition, Faculty of Food Technology, University of Agriculture in Krakow, 122 Balicka St., 30-149 Krakow, Poland; teresa.leszczynska@urk.edu.pl (T.L.); barbara.borczak@urk.edu.pl (B.B.)

[2] Department of Nutrition Technology and Consumption, Faculty of Food Technology, University of Agriculture in Krakow, 122 Balicka St., 30-149 Krakow, Poland; adam.florkiewicz@urk.edu.pl

* Correspondence: joanna.kapusta-duch@urk.edu.pl; Tel.: +48-12-662-48-16

**Abstract:** Little is still known about macro- and micronutrients in processed selected *Brassica* vegetables such as purple and green cauliflower, rutabaga and kale. This study evaluates the influence of different processing conditions (blanching and boiling) on the stability of selected macro- and micronutrients in the aforementioned vegetables. Results indicated that blanching and boiling affect the mineral content of *Brassica* vegetables. Of the examined *Brassica* vegetables, the largest losses were found for potassium and iron (on average by 39.72%).

**Keywords:** purple cauliflower; green cauliflower; rutabaga; kale; blanching; cooking; minerals

## 1. Introduction

Vegetables as a whole are considered natural sources of nutrients gifted by nature to human beings. According to the epidemiological studies, there is an opposite relationship between consumption of *Brassica* vegetables and occurrence of certain cancer forms, cardiovascular and degenerative diseases, immune dysfunction and aged-related macular degeneration. These vegetables are among the most important vegetables consumed in Europe and all over the world owing to their availability at local markets throughout the whole year, affordable cost and consumer preference. This effect is primarily due to the antioxidant nature of such plant components as, for example, products of glucosinolate decomposition, polyphenols including a group of flavonoids with anthocyanins, or phenolic acids. Moreover, vitamins A, C and E, tocopherols along with some minerals, which are present in *Brassicas*, belong to antioxidants, too [1–3].

*Brassica* vegetables contain lots of minerals, which are inorganic substances, present in all body tissues and fluids and their presence is necessary for the maintenance of certain physicochemical processes which are essential to life [4,5]. The mineral and trace element contents of plants are known to be affected by the cultivar of plant, soil conditions, weather conditions during the growing season, use of fertilizers and the state of the plant's maturity at harvest [6,7]. Minerals may be broadly classified as macro (major) or micro (trace) nutrients (elements). The macronutrients include e.g., calcium, magnesium potassium and sodium and are required in amounts greater than 100 mg/dl and the micronutrients include for example iron, copper, manganese and zinc and are required in amounts less than 100 mg/dl [8]. The concentrations of these trace minerals in vegetables may vary depending on the inherent (varieties, maturity, genetics, and age) and environmental (soils, geographical locations, season, water source and use of fertilizers) conditions of plants and animals and on methods of handling and processing [6,7]. *Brassica* vegetables are one of the most important vegetable groups grown worldwide and, compared to other vegetables,

contain a relatively large proportion of calcium. Magnesium is a macronutrient found in significant quantities in green leafy vegetables [9–11].

Essential minerals are a group of compounds that fulfil a number of functions in the human body except for being an energy source. They are a structural material, and thus components of cells, body fluids, enzymes and hormones; they participate in oxygen transfer to cells, preservation of normal neuromuscular excitability, and the water and electrolyte balance, and also help maintain the acid-base equilibrium. They are supplied to the human organism with food and water [6,12]. Calcium, the basic building material of bones and teeth, is an activator of many enzymes (including lipases and ATPases) and is also necessary for ATP (adenosine triphosphate) energy release. Furthermore, it takes part in blood coagulation and affects body balance. In addition, its ions act as an intracellular transmitter, participate in the contraction/relaxation cycles of smooth muscles and heart muscle. They also influence the functioning of cell membranes by reducing their permeability, take part in iron metabolism and interact with phosphorus, magnesium and vitamins A, D and C [13–15]. Magnesium, in turn, is a cofactor of approximately 300 enzymes, including ATPases, which determine the course of basic life processes. This micronutrient, of a crucial role for the proper structure of bone system, affects also the excitability of nervous tissue and contractility of smooth and skeletal muscles. It is also necessary for biosynthesis of proteins and nucleic acids, regulates the work of the heart and blood pressure, removes lead from the body, and activates tyrosine—the thymus hormone [16–18]. Potassium, like calcium and magnesium, plays an active role in catalysing many enzymatic reactions and controls osmotic pressure. Moreover, it maintains osmotic balance inside the cell, water/electrolyte balance, and the resting potential of the cell membrane. This mineral regulates also acid-base balance, nerve conduction, normal muscle contractility, carbohydrate transformation and synthesis of body protein [19,20]. Sodium, along with chloride ion, is involved in the regulation of body fluid osmotic pressure, transformation of nerve impulses, maintenance of acid-base balance and body protection against excessive water loss. It also controls the active transport of nutrients (vitamins, amino acids, and sugars) [21,22].

Green leafy vegetables also contain iron needed in haemoglobin formation. As regards iron, it participates in redox reactions, plays an important role in cellular respiration processes and reactions dealing which electron transport. In addition, it has a strong influence on the proper DNA biosynthesis and cell division as well as contributing to the formation of red and white blood cells. This micronutrient, occurring in some enzymes, catalyses the neurotransmitter formation from amino acids and takes part in the formation of myelin in the developing brain of young children. It is also active in the fatty acid desaturation and destruction of hydrogen peroxide. Furthermore, iron is crucial for tyrosine iodination and participates in biosynthesis of prostaglandins, catabolism of tryptophan, detoxification and immune defence of the body [23,24]. Zinc, being a component of many enzymes, is necessary for the proper metabolism of proteins, nucleic acids and carbohydrates. It accelerates the growth of hair and nails as well as wound healing. This mineral is crucial for the production of insulin and maintenance of acid-base balance. In addition, zinc ensures proper function of prostate and reproductive organs, positively affects growth and mental performance, and aids in the metabolism of vitamin A and bones as well as in transport of oxygen [25]. Copper, in turn, is necessary for haemoglobin synthesis and bone formation. This mineral is a component of the enzymes involved in intracellular respiration and is essential for the proper functioning of connective and nerve tissue. In addition, it soothes inflammatory reactions and prevents the occurrence of cardiovascular diseases. Copper, as a coenzyme, participates in redox processes and has also an important function in the synthesis of melanin, a pigment of skin and hair, playing also a large role in maintaining keratin structure [26–28]. Manganese is a micromineral required for the proper growth and development of the body and normal functioning of reproductive and nervous systems It helps to maintain proper bone structure, normal digestion and absorption of nutrients, and synthesis of haemoglobin. In addition, manganese is a component of

many enzymes that are involved in the synthesis of proteins, nucleic acids, fatty acids and, additionally, participates in cholesterol metabolism [29]. On the other hand, environmental exposure to airborne manganese is associated with neurocognitive deficits in humans. However, the bioavailability of some of these minerals might be reduced by the presence of glucosinolates, phytates and phenolics.

In most cases *Brassicas* are not eaten immediately after being harvested; hence, storage and cooking impact upon health beneficial components [3,30]. Vegetable processing is a multi-stage process. The first stage, pre-treatment, includes washing, cleaning, peeling, rinsing, grinding, and even soaking and storage in water. In the next stage, cooking is carried out by means of various methods, in different variants. The extent of mineral salt retention in vegetables depends on the applied cooking technique, the amount of water used, grind size of the raw material, as well as the form of mineral components (more or less soluble). During food processing minerals may be released from the complexes with organic compounds that may change their biological effects. Changes in mineral content in the product may result from their leaching (dissolution) and transition to broth [4].

To the best of our knowledge the majority of studies published to date have focused on other cultivars of *Brassicas*. Less information is available about other *Brassicas*, e.g., colored varieties of cauliflower (purple and green) or rutabaga.

The aim of this study was to determine the effect of blanching and cooking in water on the content of selected macronutrients (sodium, potassium, calcium, magnesium) and micronutrients (manganese, iron, zinc and copper) in fresh *Brassica* vegetables (kale, rutabaga, green (Romanesco) and purple cauliflowers), and after thermal treatment.

## 2. Materials and Methods

### 2.1. Plant Material

Four species of *Brassicas*, cultivated and eaten in Poland, were the experimental material. These were: green cauliflower—*Brassica oleracea* L. var. *botrytis* L. (Vita Verde cv.) and purple cauliflower—*Brassica oleracea* L. var. *botrytis* L. (Graffiti cv.); kale—*Brassica oleracea* var. *acephala* (Winterbor $F_1$ cv.) and rutabaga—*Brassica napus* L. var. *napobrassica* (Wilhelmburska cv.). The vegetables originated from the "Polan" Plant and Horticultural Seed Production Centre Ltd. Krakow (Poland) and the "Traf" Producers' Cooperative in Tropiszów (Poland).

Immediately after harvest, vegetable samples were prepared for analyses. From a 5 kg batch of every vegetable, samples of usable parts were taken, cut along the axis in a ratio of 1:4 or 1:8 (depending on size), washed under ultrapure water and dried on filter paper. Afterwards, the samples were shredded mechanically, frozen at $-22\,^{\circ}C$, and then freeze-dried in a Christ Alpha 1–4 apparatus (Christ, Frankfur, Germany). In addition, the freeze dried material was comminuted in a Knifetec 1095 Sample Mill (Tecator, Hoganas, Sweden) until a homogenous sample with the smallest possible particle diameter was obtained. The process of blanching was conducted in a stainless steel pot. Vegetables were immersed in water at 96–98 $^{\circ}C$ for approximately 3 min, immediately chilled by cold water, and finally drained.

Another vegetable batch was cooked simultaneously in the traditional way. A 15-min cooking was performed in a stainless steel pot on an electric hob. Vegetables were cooked in ultrapure water and initially without a lid, according to the principle "from farm to fork". The proportion of water to the raw material was 5:1 (*w/w*). Afterwards, the boiled vegetables were prepared as previously described for fresh vegetables.

### 2.2. Analytical Methods

Vegetable samples were examined for the dry matter content by drying at 105 $^{\circ}C$, under normal pressure conditions, according to PN-90/A-75101/03 [31]. The principle of the applied method was a decrease in mass upon removal of water from the product over drying.

The contents of mineral compounds such as calcium, magnesium, potassium, and sodium, were established using the validated atomic absorption spectrometry method with the atomization in a flame (FAAS, Varian AA240FS of the Varian Company, Palo Alto, CA, USA) in accordance with the Polish Standard PN-EN-15505:2009 [32], while for iron, manganese, zinc, and cuprum contents, the Polish Standard PN-EN 14084:2004 [33], was applied. Samples were mineralized with 65% nitric acid (Suprapur, MERCK, Kenilworth, NJ, USA, cat. no 1.00441), in the amount of 10 mL/0.5 g sample. The process was carried out in Teflon containers at max. temperature of 200 °C for 40 min by means of a high pressure microwave method (MarsXPres, CEM, Stallings, NC, USA). Potassium and sodium were determined by adding buffer solution of Schuhknecht and Schinkel (caesium chlorate and aluminum nitrate in the respective concentrations of 50 g/L and 250 g/L; MERCK, kat. no 102037), while the content of calcium and magnesium, by adding buffer solution of Schinkel (caesium chloride and lanthanum chloride in the concentrations of 10g/L; MERCK, kat. no 1.16755). Minerals were determined at the following wavelengths: K—766.5 nm, Ca—222.7 nm, Mg—285.2 nm, Na—589.0, Zn—213.9 nm, and Mn—279.5 nm. Certified reference material NCS ZC73012—GSB-5 (China National Analysis Centre for Iron and Steel, Beijing, China) was used to validate the accuracy of applied methods. Methods were fully validated and checked by internal quality control procedure PN-EN 13804:2013-06, 2013 [34] as well as interlaboratory/proficiency tests.

### 2.3. Statistical Analysis

Technological treatments and analyses were conducted in triplicate. Data were expressed as means ± SD (standard deviation). Determination of significance of differences between means by the post-hoc Duncan's test were performed at a significance level of $p \leq 0.05$, using the Statistica 12 software package (StatSoft Inc., Tulsa, OK, USA).

### 3. Results

As the dry matter content in the vegetable varies depending on the process applied, all the results presented below along with conclusions have been discussed based on the results calculated per the dry matter unit taking into account the efficiency of the process.

The process efficiency was calculated (expressed as percentage) by dividing mass of processed material/sample by mass of fresh sample. Reduction of mineral compounds content (expressed as a percent-age) was calculated as the difference between their absolute content in the material after the treatment and the absolute content in the raw material, taking into account the process efficiency. In consequence, only an effect of the process applied was shown.

### 3.1. Macronutrients

Blanching led to a significant fall ($p \leq 0.05$) in potassium (K) content only in kale (by 24.0%), when compared with the raw vegetables. With regard to other vegetables, the content of this macromineral decreased insignificantly ($p > 0.05$), on average, by 5.5%. A similar observation was made for sodium (Na) content, which was reduced due to the technological treatment in all discussed *Brassica* vegetables, compared to raw vegetables; however, significant reductions ($p \leq 0.05$) were only for kale (by 7.6%) and purple cauliflower (by 17.4). The process of blanching caused a significant decrease in calcium content in majority of the examined *Brassica* vegetables compared to raw ones, which was 6.3 (kale), 7.5 (rutabaga), and 10.3% (purple cauliflower).

As for the last discussed micronutrient, magnesium (Mg), its reduction due to the hydrothermal treatment was significant ($p \leq 0.05$) only in rutabaga, compared to the fresh vegetable, and was 5.4%. In general, cooking led to significant ($p \leq 0.05$) decrease in the content of this micronutrients. With regard to K and Na, losses were respectively: 40.4% and 21.8, in kale; 29.6 and 39.4, in rutabaga; 45.9 and 46.6, in green cauliflower; and 41.6 and 26.3%, in purple cauliflower, compared to the fresh vegetable. In the analysed *Brassica*

vegetables, the largest losses resulting from cooking, on average of 39.4%, were observed for potassium.

Cooking reduced significantly ($p \leq 0.05$) the level of Ca and Mg in three of the four examined *Brassicas* compared to raw vegetables. The losses in calcium content were: 20.2%, in kale; 30.2% in rutabaga, and 19.2% in purple cauliflower, while in magnesium content these were: 15.3% in rutabaga, 28.2%, in green cauliflower, and 40.9% in purple cauliflower. Interestingly, calcium content in green cauliflower increased significantly ($p \leq 0.05$) by 18.0%, compared to the raw vegetable.

### 3.2. Micronutrients

In turn, blanching led to a significant ($p \leq 0.05$) decrease in the contents of Cu, Mn and Fe by respectively: 4.7%; 13.5% and 12.8% in kale; and 16.3%; 5.5% and 2.2% in rutabaga, compared to raw vegetables. In both vegetables, blanching had no effect ($p > 0.05$) on zinc content. In both cauliflowers having undergone blanching the content of copper decreased significantly ($p \leq 0.05$), on average, by 12.3%, while iron and zinc content dropped only in green cauliflower, on average, by 12.5%, compared to raw vegetables.

In general, cooking reduced significantly ($p \leq 0.05$) the level of these micronutrients; losses in Cu and Fe were respectively 12.9% and 48.4% (kale); 36.4% and 16.8% (rutabaga); 36.6% and 49.4% (green cauliflower), and 27.1% and 22.0% (purple cauliflower), compared to raw vegetables. This process lowered significantly ($p \leq 0.05$) also manganese content in kale, rutabaga and purple cauliflower, by respectively 27.6% 14.2% and 44.6%, compared to the raw vegetable. In green cauliflower, there was a significant 16.4% rise in the amount of manganese compared to raw vegetables. The content of zinc decreased significantly ($p \leq 0.05$) in two out of four analysed vegetables, i.e., in kale and green cauliflower, by 49.8% and 25.8%, respectively, compared to raw vegetable. In all investigated vegetables, the largest total losses of microminerals due to cooking, on average by 35.0%, were observed for iron.

## 4. Discussion

### 4.1. Macronutrients

The contents of macronutrients (potassium, sodium, calcium, and magnesium) in raw and hydrothermally processed vegetables immediately after blanching and cooking, are summarized in Table 1.

As it happens, most food composition tables have been prepared in temperate climate countries. Differences in climate, soil, agronomic practices and plant cultivar are the probable causes for differences in nutrient levels and indicate the need for locally prepared food composition [6].

The highest content of potassium was recorded in raw kale (33,497.01 mg/kg dm) and lowest in rutabaga (20,953.65 mg/kg dm). The content of this macronutrient in both cauliflowers (green and purple rose) was similar. It should be noted that individual vegetables significantly differed in the content of this ingredient, which was strongly dependent on environmental, cultivating or genetic factors [7,9]. Similar or lower levels of potassium were identified by other authors in *Brassica* [4,9,35–37]. A much higher than the result obtained in this study received a team of Baloch et al. [38] and Mansour et al. [39] in cauliflower. However, Martínez et al. [40], obtained a slightly higher result for the content of the discussed macronutrient in swedes, compared to that obtained in this study.

**Table 1.** The content (mean ± SD) of selected minerals in kale, rutabaga, green and purple cauliflower subjected to different technological processing (mg/kg dry matter). The values in the same columns denoted with different letters: a, b, c differ statistically significantly at $p \leq 0.05$. * r-raw; b-blanched, c-cooked.

| Vegetables | | Potassium | Sodium | Calcium | Magnesium | Copper | Manganese | Iron | Zinc |
|---|---|---|---|---|---|---|---|---|---|
| kale | r * | 33,497.01 [c] ± 22.66 | 364.56 [c] ± 6.75 | 15,968.12 [c] ± 333.52 | 1565.98 [a] ± 5.66 | 2.74 [c] ± 0.15 | 22.46 [c] ± 0.96 | 161.37 [c] ± 0.90 | 67.86 [b] ± 1.49 |
| | b * | 25,671.11 [b] ± 244.64 | 331.04 [b] ± 15.74 | 15,087.78 [b] ± 47.18 | 1558.61 [a] ± 0.94 | 2.63 [b] ± 0.03 | 19.59 [b] ± 0.35 | 141.93 [b] ± 4.77 | 64.89 [b] ± 1.78 |
| | c * | 20,977.63 [a] ± 816.47 | 291.74 [a] ± 74.00 | 13,392.38 [a] ± 197.72 | 1533.10 [a] ± 53.28 | 2.51 [a] ± 0.03 | 17.08 [a] ± 0.13 | 87.51 [a] ± 0.89 | 35.76 [a] ± 0.74 |
| rutabaga | r | 20,953.65 [b] ± 429.36 | 441.35 [b] ± 19.35 | 3394.43 [c] ± 86.28 | 1129.05 [c] ± 10.43 | 2.79 [c] ± 0.07 | 8.15 [c] ± 0.42 | 250.75 [c] ± 1.43 | 18.58 [a] ± 0.33 |
| | b | 20,013.68 [b] ± 180.13 | 424.21 [b] ± 2.98 | 3188.42 [b] ± 40.18 | 1083.68 [b] ± 6.70 | 2.37 [b] ± 0.07 | 7.81 [b] ± 0.08 | 248.95 [b] ± 3.72 | 18.74 [a] ± 0.45 |
| | c | 15,321.81 [a] ± 334.63 | 277.64 [a] ± 18.75 | 2461.31 [a] ± 25.50 | 993.43 [a] ± 19.82 | 1.84 [a] ± 0.09 | 7.26 [a] ± 0.08 | 216.65 [a] ± 0.55 | 17.35 [a] ± 0.47 |
| green cauliflower | r | 31,985.46 [b] ± 921.68 | 760.90 [b] ± 14.22 | 3789.97 [a] ± 14.35 | 1366.79 [b] ± 11.95 | 3.16 [b] ± 0.15 | 17.64 [a] ± 0.46 | 69.70 [c] ± 2.45 | 56.09 [b] ± 0.89 |
| | b | 31,066.28 [b] ± 647.62 | 754.17 [b] ± 5.41 | 3790.86 [a] ± 14.96 | 1351.68 [b] ± 12.72 | 2.73 [a] ± 0.08 | 17.32 [a] ± 0.84 | 58.54 [b] ± 4.80 | 53.88 [a] ± 2.17 |
| | c | 22,767.62 [a] ± 781.97 | 534.18 [a] ± 19.02 | 4086.45 [b] ± 180.09 | 1290.73 [a] ± 25.89 | 2.63 [a] ± 0.15 | 19.41 [b] ± 0.32 | 46.38 [a] ± 0.67 | 54.72 [a] ± 1.52 |
| purple cauliflower | r | 29,681,87 [b] ± 622,98 | 847.29 [b] ± 31.12 | 3797.75 [c] ± 24.60 | 1671.95 [b] ± 2.53 | 2.95 [c] ± 0.15 | 20.76 [b] ± 0.12 | 60.23 [b] ± 0.94 | 39.87 [a] ± 2.39 |
| | b | 28,620.92 [b] ± 183.65 | 710.10 [a] ± 15.91 | 3458.97 [a] ± 45.37 | 1607.99 [b] ± 2.31 | 2.74 [b] ± 0.06 | 20.17 [b] ± 0.12 | 59.00 [b] ± 1.67 | 38.94 [a] ± 1.09 |
| | c | 20,785.68 [a] ± 323.74 | 749.08 [a] ± 21.55 | 3690.02 [b] ± 12.84 | 1186.25 [a] ± 57.90 | 2.58 [a] ± 0.22 | 13.81 [a] ± 0.21 | 56.38 [a] ± 0.88 | 38.08 [a] ± 0.67 |

The greatest potassium reduction was observed in traditionally cooked *Brassica* (kale and both types of cauliflower) and the lowest, as was expected, in blanched vegetables (rutabaga, green and purple cauliflower) (Table 2). The behavior of minerals during blanching is related to their solubility. Potassium, the most abundant mineral in vegetables, is extremely mobile and is easily lost by leaching during blanching because of its high solubility in water. Calcium and magnesium are generally bound to the plant tissue and are not readily lost by leaching and sometimes can even be taken up by vegetables during blanching from the processing water in areas with hard water [36]. Similar observations were made by other authors [4,36,39] on traditionally cooked and steamed cauliflower, broccoli and Brussels sprouts. Sikora and Bodziarczyk [37], after 12–15 min of cooking, observed a decrease in the content of this micronutrient in kale by 38%, which is almost congruent with our results.

**Table 2.** Losses of selected minerals in in kale, rutabaga, green and purple cauliflower subjected to different technological processing (%) *.

|  |  | Potassium | Sodium | Calcium | Magnesium | Copper | Manganese | Iron | Zinc |
|---|---|---|---|---|---|---|---|---|---|
| kale | blanched | 24.0 | 7.6 | 6.3 | 1.3 | 4.7 | 13.5 | 12.8 | 5.5 |
|  | cooked | 40.4 | 21.8 | 20.2 | 6.8 | 12.9 | 27.6 | 48.4 | 49.8 |
| rutabaga | blanched | 5.9 | 5,3 | 7.5 | 5.4 | 16.3 | 5.5 | 2.2 | 0.6 |
|  | cooked | 29.6 | 39.4 | 30.2 | 15.3 | 36.4 | 14.2 | 16.8 | 10.1 |
| green cauliflower | blanched | 5.6 | 3.6 | 2.8 | 3.9 | 16.1 | 4.5 | 18.4 | 6.6 |
|  | cooked | 45.9 | 46.6 | 18.0 | 28.2 | 36.6 | 16.4 | 49.4 | 25.8 |
| purple cauliflower | blanched | 5.0 | 17.4 | 10.3 | 5.3 | 8.5 | 4.3 | 3.5 | 3.8 |
|  | cooked | 41.6 | 26.3 | 19.0 | 40.9 | 27.1 | 44.6 | 22.0 | 20.4 |

* Percentage losses of compounds determined by the effects of hydrothermal processing were calculated in compliance with mass balance changes. This way of presenting data eliminated the influence of thinning the content of the element (with water), and demonstrated only the influence of the processing technique applied.

Minerals compared with vitamins are more resistant to food processing. However, these components cannot tolerate alterations after exposure to light, moisture, heat or oxygen during the processing and storage of food materials. Moreover, processing can also improve accessibility of micronutrients by decreasing antinutrient levels [40]. The highest content of sodium in fresh material was noted in purple cauliflower (847.29 mg/kg dm) and lowest in kale (364.56 mg/kg dm). The results reported by the Florkiewicz and Berski [4] for *Romanesco* green cauliflower were slightly higher, in comparison to those obtained in this study (760.90 mg/kg dm), while for white floret cauliflower they were much higher. This highlights even more the diversity in the content of mineral components depending on many factors, such as variety, agrotechnical factors and climate conditions. The iron contents in various *Brassica* species, reported by other authors, were much higher [35–37,39]. In cauliflowers grown in Bangladesh, the sodium content was significantly differentiated: 165.73 mg/100 g dm for naturally grown and 429.11 mg/100 g dm watered with waste water [9]. Blanching and cooking resulted in significant losses of this micromineral, which agrees with the findings of other authors [4,36,39]. The largest losses of this component under the influence of the cooking process were observed in green cauliflower (46.6%) which is rich in this ingredient, and the smallest in the kale (21.8%).

*Brassica* plants have been found to be rich in many minerals including calcium. All analyzed vegetables contained over 3300.00 mg/kg of calcium. Among analyzed *Brassica* the highest content was observed for kale (15,968.98 mg/kg dm) and the lowest for rutabaga (3394.43 mg/kg dm) (Table 1). Interestingly, kale and cauliflower exhibit excellent calcium bioavailability. Comparing the determined calcium content of the analyzed fresh *Brassica* vegetables with literature data, it could be concluded that they were quite similar or higher [4,36–41]. Results of calcium content analyses in cauliflowers originating from Bangladesh were significantly higher (730.38 mg/100 g dm) [35]. The content of calcium in rutabaga, given by Martínez et al. [41], was much higher compared to our results.

Blanching did not reduce significantly the content of this compound only in green cauliflower. In all vegetables mean losses amounted to 5.15% compared to raw vegetables. Cooking, presumably, contributed to much higher and significant calcium losses, on average, by about 22%, the highest losses, by 30.2%, were found for rutabaga. Other authors also observed calcium losses in *Brassica* vegetables during various types of thermal process [4,36,37,39].

The content of magnesium in raw *Brassica* vegetables varied from 1129.05 (rutabaga) to 1671.95 mg/kg dm (purple cauliflower). Higher levels of this macroelement were found by other researchers [35,36,39,40]. Slightly higher amounts of magnesium were identified in the cauliflower by Baloch et al. [38] and Florkiewicz and Berski [4]. Miller-Cebert et al. [41] stated twice higher magnesium content in fresh kale, while Martínez et al. [40] noted the same for rutabaga. Cooking reduced significantly the content of this macromineral in all examined *Brassica* vegetables, except for kale, in which losses amounted to only 6.8%. These results are congruent with the findings of Florkiewicz and Berski [4], who observed its losses in all examined *Brassica* vegetables due to conventional cooking in water.

*4.2. Micronutrients*

Microelements are chemical elements that occur in very small quantities in plant and animal organisms. Their contents (copper, manganese, iron and zinc) in raw and hydrothermally processed vegetables immediately after blanching and cooking, are summarized in Table 1.

The copper content in fresh vegetables ranged from 2.74 (kale) to 3.16 (green cauliflower) mg/kg dm, which indicates small variation in content of this element in fresh material (Table 1). In comparison with our results, most authors report either much lower [40], or much higher magnesium contents in *Brassica* vegetables [4,35,37,41].

Both blanching and cooking led to a significant reduction of this microelement. The finding was confirmed by Florkiewicz and Berski [4], but only with regard to two examined brassica vegetables (green cauliflower and broccoli), which underwent conventional cooking. In turn, Sikora and Bodziarczyk [38] observed a 12% increase of this component compared to the kale untreated hydrothermally.

The manganese content in fresh vegetables ranged from 8.15 (rutabaga) to 22.46 (kale) mg/kg dm, that indicates on high variation in content of this element in fresh material depending on the species (Table 1). Higher results regarding manganese content in *Brassica* could be found in the literature [35–37]. Ahmed and Ali [36] found no statistical difference between manganese content in fresh cauliflower and water or steam boiled. This was consistent with the findings of Sikora and Bodziarczyk [37] and Florkiewicz and Berski [4], who only in cooked broccoli observed a significant decrease in magnesium content, by 41.3% compared to the fresh vegetable. This work showed that cooking led to significant losses of this microelement in most of the analysed vegetables; the only exception was boiled green cauliflower, in which its content increased significantly, by 16.4%.

The human body shows a varied demand for iron, which is dependent on age, sex and physical condition. However, it should be taken into consideration that iron assimilation is approximately 5% for non-animal products. By analyzing the data in Table 1, the highest iron content in raw material was recorded in rutabaga (250.75 mg/kg dm) and the lowest in purple cauliflower (60.23 mg/kg dm). Much higher values than given in this paper were published by Ahmed and Ali, [36], Miller-Cebert et al. [35], Baloch et al. [38]. On the other hand, as reported by Ayaz et al. [40], Martínez et al. [41] and Mansour et al. [39], its levels in kale, rutabaga and cauliflower were significantly lower compared to our results. In turn, Florkiewicz and Berski [4] stated percentage reduction of magnesium depending on the applied processing, the highest for conventional cooking in water, 13.8% (*Romanesco* cauliflower) and 38.2% (broccoli). In contrast, Ahmed and Ali [36] observed no differences in iron content between fresh cauliflower and that blanched and cooked by means of various methods.

The zinc content of the analyzed samples varies enormously depending on the species, ranging from 18.58 (rutabaga) to 67.86 mg/kg dm (kale) (Table 1). With regard to zinc content, as was in the case of other aforementioned mineral components, the results obtained by other authors were lower [37,39,40] or higher [35,36,41], compared to our findings.

This study showed a significant fall in zinc content due to cooking in green cauliflower and kale, compared to raw vegetables. Similar losses of 13% in kale and 23% in cauliflower were reported by Sikora and Bodziarczyk [37] and Mansour et al. [39]. On the other hand, Ahmed and Ali [36] found no significant changes in the content of this micronutrient due to the applied hydrothermal processes, which is congruent with our results for rutabaga and purple cauliflower.

## 5. Conclusions

*Brassica* species are the good sources of valuable nutrients, including minerals. The mineral contents can vary significantly depending on species, variety, plant part or environmental factors. Their content is strongly influenced by methods of preparation. As shown in this study, blanching and boiling affect the minerals content of *Brassica* vegetables. Of the examined *Brassica* vegetables, the largest losses, on average by 39.4%, were found for potassium. With regard to microcomponents present in all examined vegetables having undergone cooking, the largest declines were in iron, by 35.0% on average.

**Author Contributions:** J.K.-D. conceived and designed the study, participated in data collection and analyses and drafted the manuscript. A.F. analysed results and drafted the manuscript. B.B. participated in data collection and analyses and drafted the manuscript. T.L. contributed equally to the literature review and analysis of results. B.B. and T.L. contributed to the design and supervision of the study, and edited the manuscript. A.F. contributed to the data collection, generation of tables and writing of the manuscript. All authors have read and agreed to the published version of the manuscript.

**Funding:** This research received no external funding.

**Institutional Review Board Statement:** Not applicable.

**Informed Consent Statement:** Not applicable.

**Data Availability Statement:** Not applicable.

**Conflicts of Interest:** The authors declare no conflict of interest.

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
