# Peer review of "Directions of Changes in the Content of Selected Macro- and Micronutrients of Kale, Rutabaga, Green and Purple Cauliflower Due to Hydrothermal Treatment"

_applsci, doi:10.3390/app11083452_

Round 1

Reviewer 1 Report

185-193 repeated text from 175-183, remove it 

368 preparation

Author Response

Please see the attachment. There is a revised version of the work with marked two amendments (line no 386 and removing the paragraph from the verses 185-193).

Reviewer 2 Report

The manuscript "Directions of Changes in the Content of Selected Macro- and Micronutrients of Kale, Rutabaga, Green and Purple Cauliflower Due to Hydrothermal Treatment" is well written and organised, but the main weak point is the limited originality. Standard methods and techniques are used and in this sense the scientific advance is quite limited. Nevertheless, results can be interesting for a sector of readers.

Several points that should be considered before its acceptation for publication in "Applied Sciences" are as follows:

1.- Several mistakes have been detected in the manuscript:

  • Line 16: please, change the percentage to 39.72%
  • Line 145: Analitycal --> Analytical
  • Line 157: 200 8C --> 200 ºC
  • Line 159: cesium --> caesium
  • Line 171: means 6 SD --> means ± SD
  • Table 1 & 2: please, replace all "," by "."
  • Line 255: dm -> dry matter to avoid confusion with decimeter
  • Line 269: generalny --> generally

2.- Please, arrange table 1 in order to have values ± SD in the same line to improve readability.

3.- In line 131, authors mention that samples are washed under tap water. Despite it is unrealistic for normal uses, why not with distilled water to avoid possible transfer of cations from the tap water? The possible contribution is mentioned along the manuscript, but the effect could be taken exactly taken into account. Blanks of tap water, or the determination of the contents of cations in tap water have been considered? If not, an independent analysis of the tap water shoud be carried out and results showed in the manuscript. It is clear that the composition can change from day to day, but at least can serve as reference for a more accurate discussion about e.g. magnesium or calcium, where increase of concentration is reported when subjecting samples to thermal treatments.

4.- Lines 175-183 are exactly repeated in lines 185-193. Please, arrange it by deleting one of such paragraphs.

Author Response

Response to Reviewer’s (#2) Comments

Dear Reviewer,

please see the attachment (the corrected version of manusript).

  1. Some mistakes have been corrected in the paper:

   Line 16: please, change the percentage to 39.72% DONE

    Line 145: Analitycal --> Analytical CORRECTED

    Line 157: 200 8C --> 200 ºC CORRECTED

    Line 159: cesium --> caesium CORRECTED

    Line 171: means 6 SD --> means ± SD DONE

    Table 1 & 2: please, replace all "," by "." DONE

    Line 255: dm -> dry matter to avoid confusion with decimeter DONE

    Line 269: generalny --> generally CORRECTED

  1. Please, arrange table 1 in order to have values ± SD in the same line to improve readability.

IT HAS BEEN DONE.

  1. In line 131, authors mention that samples are washed under tap water. Despite it is unrealistic for normal uses, why not with distilled water to avoid possible transfer of cations from the tap water? The possible contribution is mentioned along the manuscript, but the effect could be taken exactly taken into account. Blanks of tap water, or the determination of the contents of cations in tap water have been considered? If not, an independent analysis of the tap water shoud be carried out and results showed in the manuscript. It is clear that the composition can change from day to day, but at least can serve as reference for a more accurate discussion about e.g. magnesium or calcium, where increase of concentration is reported when subjecting samples to thermal treatments.   

The part of Materials and Methods has been improved according to the best knowledge of the authors of the manuscript. Please, see attached manuscript after revision.

Explanation:

After consultations with with other Co-authors of this publication, who also carried out the blanching and cooking process as well as the pre-treatment of vegetables, it turned out that all these processes were carried out using ultrapure water. Relevant corrections have been made in the material and methods section. Additionally, the study showed strong decreases in both calcium and magnesium in each vegetable under the influence of technological processing. Please refer to Table 2 showing only the losses in the content of micro- and macronutrients due to the blanching and cooking processes.

  1. Lines 175-183 are exactly repeated in lines 185-193. Please, arrange it by deleting one of such paragraphs.

IT HAS BEEN DONE.

English language and style are fine/minor spell check required 

The manuscript was checked again for the spelling of the English language. 

The authors are very grateful to the anonymous Referees for their valuable comments.
